# The Conceptual Independence of Health Status, Respiratory Symptoms and Dyspnea in Chronic Obstructive Pulmonary Disease in Real Clinical Practice

**DOI:** 10.3390/diagnostics13152492

**Published:** 2023-07-27

**Authors:** Koichi Nishimura, Masaaki Kusunose, Mio Mori, Ayumi Shibayama, Kazuhito Nakayasu

**Affiliations:** 1Visiting Researcher, National Center for Geriatrics and Gerontology, 7-430, Morioka-cho, Obu 474-8511, Japan; 2Clinic Nishimura, 4-3. Kohigashi, Kuri-cho, Ayabe 623-0222, Japan; 3Department of Respiratory Medicine, National Center for Geriatrics and Gerontology, 7-430, Morioka-cho, Obu 474-8511, Japan; kusunose@ncgg.go.jp (M.K.); mio-mori@ncgg.go.jp (M.M.); 4Department of Nursing, National Center for Geriatrics and Gerontology, 7-430, Morioka-cho, Obu 474-8511, Japan; ayuminarita3@gmail.com; 5Data Research Section, Kondo P.P. Inc., 17-25, Shimizudani-cho, Tennoujiku, Osaka 543-0011, Japan; nakayasu@mydo-kond.co.jp

**Keywords:** COPD, dyspnea, health status, respiratory symptoms, surveys and questionnaires

## Abstract

The hypothesis that health status is the highest ranking concept, followed by respiratory symptoms and dyspnea as the lowest ranking concepts in subjects with chronic obstructive pulmonary disease (COPD) was tested in a real clinical setting with 157 subjects with stable COPD. Spearman’s rank correlation coefficients for scores of health status using the COPD Assessment Test (CAT), respiratory symptoms using the COPD Evaluating Respiratory Symptoms (E-RS) and dyspnea using Dyspnea-12 (D-12) between any two were 0.6 to 0.7. Upon categorizing the patients as “abnormal” or “normal” according to the threshold, it was found that 30 patients (19.1%) had dyspnea, respiratory symptoms and impaired health status. Dyspnea was considered an important part of respiratory symptoms, though seven patients had dyspnea but no respiratory symptoms. There were 10 patients who had respiratory symptoms without dyspnea but without health status problems. Furthermore, there were six patients who had both dyspnea and respiratory symptoms but whose health status was classified as fine. Thus, the hypothesis was correct in approximately 85% of cases.

## 1. Introduction

During the last two decades, patient-reported outcomes (PROs) have been considered important in the evaluation of healthcare services or as a primary or secondary endpoint of clinical trials in the treatment of patients with chronic obstructive pulmonary disease (COPD) [1,2]. As several tools have been reported in the literature, it may be difficult to understand how the conceptual framework from which each instrument derives differs between tools. Jones et al. developed the St. George’s Respiratory Questionnaire (SGRQ) and the COPD Assessment Test (CAT) for health status measurements in subjects with COPD [3,4,5,6,7]. Yorke et al. reported that Dyspnea-12 (D-12) provides a global score of the severity of breathlessness and can measure dyspnea in a variety of diseases [8,9,10]. However, as dyspnea is obviously different from health status, it would not be easy to explain how dyspnea is relevant to health status [11].

The Global Initiative for Chronic Obstructive Lung Disease (GOLD) launched a classification system in 2011 [12] called the revised “combined COPD assessment” classification, in which symptoms should be evaluated using the modified Medical Research Council (mMRC) dyspnea scale or the CAT. However, as the former is regarded as a tool for measuring dyspnea and the latter as a health status measure, the results may differ somewhat from what the symptoms really are, and they may be a little different from what the symptom should be. The discrepancy between the mMRC and CAT scores has since been widely debated [13,14,15,16,17,18,19,20]. Although some have attributed the discordance to the sensitivity of the tools’ measurement properties, it may not be surprising, as dyspnea and health status differ conceptually.

On the other hand, Leidy et al. created a reliable and valid instrument for evaluating the severity of respiratory symptoms in stable COPD using 11 respiratory symptoms items from the 14-item Exacerbations of Chronic Pulmonary Disease Tool Patient-Reported Outcome (known as EXACT-PRO) [21,22,23,24]. This is the Evaluating Respiratory Symptoms in COPD (E-RS) [23,24], which was designed as a daily diary to be easily administered by clinical study subjects using a personal digital assistant or smartphone. The original developers of the CAT, D-12 and E-RS mentioned that they derive from different conceptual frameworks, but theoretically, dyspnea may be included in respiratory symptoms, and this symptom may be one of the essential components of health status. Unfortunately, however, they are often undifferentiated and are used almost interchangeably in everyday clinical practice.

The GOLD states that chronic and progressive dyspnea is the most characteristic symptom of COPD, and cough with sputum production is present in up to 30% of patients. Hajiro et al. reported that using stepwise multiple regression analyses, the Baseline Dyspnea Index (BDI) score, anxiety using the Hospital Anxiety Depression Scale (HAD) and maximal oxygen uptake (O_2_max) accounted for 61% of the variance in the SGRQ [11]. Therefore, health status is the highest level concept, dyspnea is the lowest level concept, and respiratory symptoms are in between, which is the hypothesis that this study seeks to test. This hypothesis is depicted in Figure 1, in which dyspnea would be reflected in respiratory symptoms and respiratory symptoms in the health status, as it can be commonly accepted that breathlessness is included in respiratory symptoms and that this symptom is one of the essential components of health status in subjects with COPD. This may be helpful in solving the question of whether or not they can be used interchangeably in subjects with COPD in clinical practice.

The aim of the present study was to ascertain whether the conceptual independence of health status, respiratory symptoms and dyspnea is maintained in the clinical practice of COPD. For this purpose, the authors examined whether the distribution of cases is consistent with the hypothesis when the cases are classified as “abnormal” or “normal” or “with (disability)” or “without (disability)” according to the respective thresholds based on the scores of the evaluation tools for the three concepts.

## 2. Materials and Methods

### 2.1. Participants

We recruited 157 consecutive patients with stable COPD who attended the outpatient clinic in the Department of Respiratory Medicine of the National Center for Geriatrics and Gerontology (NCGG) from September 2013 to February 2022. Inclusion criteria included being over 50 years of age, having a smoking history of more than 10 pack-years, having chronic fixed airflow limitation, attending the clinic regularly for more than half a year, having no uncontrolled comorbidities and having no variation in treatment in the preceding four weeks. Chronic fixed airflow limitation was defined as a maximum ratio of forced expiratory volume in 1 s (FEV_1_) to forced vital capacity (FVC) of less than 0.7. All participants provided written informed consent, and the research was approved by the National Center for Geriatrics and Gerontology Institute’s Ethics Committee (No. 1138-3).

### 2.2. Measurements

Baseline pulmonary function measurements of the participants were taken on a single day, which included postbronchodilator spirometry (CHESTAC-8800; Chest, Tokyo, Japan), residual volume (RV) measured using the closed-circuit helium method and diffusing capacity for carbon monoxide (DL_CO_) assessed using the single-breath technique as reported by the American Thoracic Society and European Respiratory Society Task Force in 2005 [25]. Calculations of the predicted values for FEV_1_ and vital capacity were carried out as recommended by the Japan Respiratory Society [26].

### 2.3. Patient-Reported Measurements

Validated Japanese versions of the following patient-reported outcome measurement tools were used in the present study: the CAT to measure health status, D-12 to assess the severity of breathlessness and E-RS to analyze and quantify respiratory symptoms [27,28]. The St. George’s Respiratory Questionnaire (SGRQ) (version 2) and the Hyland Scale were also administered as a standard procedure. The former consists of 50 items divided into the three components of symptoms, activity and impact, and a total score ranging from 0 to 100 is calculated [3,11]. Higher scores on the SGRQ indicate a more severe state of health. The latter is a global health scale with scores ranging from 0 to 100, where 0 = “might as well be dead” and 100 = “perfect quality of life” [29,30].

To assess the severity of dyspnea, we used D-12, which consists of twelve items (seven physical and five affective), each with a four-point grading scale (0–3), producing a total score (range 0–36, with higher scores representing more severe breathlessness) [8,9,10,28]. CAT is a questionnaire consisting of eight items scored from 0 to 5 in relation to cough, phlegm, chest tightness, breathlessness going up hills/stairs, activity limitations at home, confidence leaving home, sleep and energy [5,6,7,27]. CAT scores range from 0 to 40, with a score of zero indicating no impairment. The E-RS uses 11 respiratory symptom items from the 14-item EXACT-PRO, where scores range from 0 to 40, with higher scores indicating more severe symptoms [23,24]. The E-RS Total score represents the severity of the general respiratory symptoms. Three subscales were also used in this analysis. A Japanese translation was created and provided by the original developers, and they recommended using an electronic version to collect the answers. However, no electronic devices with the Japanese version of the EXACT-PRO or E-RS were available, so all surveys were conducted using a paper-based method.

We reported in 2019 that from the data obtained from 646 healthy nonsmoking subjects, the reference values for the D-12, E-RS Total and CAT scores were considered to be ≤1, ≤4 and ≤9, respectively. These reference values were also used as the thresholds in the present study [31].

### 2.4. Statistical Methods

Score distributions of the tools were evaluated using the Shapiro–Wilk test and through inspection of histograms. Spearman’s rank correlation tests were used to examine relationships between two sets of data. The significance of between-group differences was determined using Kruskal–Wallis test and Steel–Dwass test. The relationships between three groups were also analyzed using a Venn diagram. All *p*-values less than 0.05 were deemed to be statistically significant. The results are expressed as mean ± standard deviation (SD), with some exceptions in the tables.

## 3. Results

### 3.1. Subject Characteristics

A total of 157 consecutive patients (144 men) with COPD and a wide range of FEV_1_ (69.8 ± 20.4% pred.) participated. In total, 120 subjects were former smokers, while 37 were current smokers. Their demographic details, as well as the results of the pulmonary function tests, are listed in Table 1. Using the classification of the severity of airflow limitation of the Global Initiative for Chronic Obstructive Lung Disease (GOLD) criteria, 53 subjects (33.8%) were in Stage 1 (defined as FEV_1_ ≧ 80% predicted), 79 (50.3%) in Stage 2 (50% ≦ FEV_1_ < 80% predicted), 18 (11.5%) in Stage 3 (30% ≦ FEV_1_ < 50% predicted) and 7 (4.5%) in Stage 4 (FEV_1_ < 30% predicted) (Table 2). Relatively few patients with severe or very severe COPD were involved in the present study.

### 3.2. Distribution of Scores and Correlation between Tools

The hypothesis that the scores obtained are normally distributed was rejected in the D-12, CAT and E-RS, including their subscales, as shown in Table 3 (Shapiro–Wilk test, all *p* < 0.001). They were skewed toward the milder ends, and a floor effect was observed in all scores. This effect was most pronounced for D-12 (47.8%) and the least for the CAT (5.1%).

Regarding the inter-relationships between the D-12 Total, CAT and E-RS Total scores, they were significantly correlated with each other (D-12 Total vs. CAT, Spearman’s correlation coefficient (Rs) = 0.603, *p* < 0.001; D-12 Total vs. E-RS Total, Rs = 0.655, *p* < 0.001; and CAT vs. E-RS Total, Rs = 0.675, *p* < 0.001) (Table 1). All of the correlation coefficients were below 0.7 or what is occasionally regarded as the level suggestive of conceptual equivalence. Scatterplots showing the relationships between tools are depicted in Figure 2.

The correlation coefficients between the baseline characteristics and each PRO measure score are presented in Table 1. The correlation coefficients between the D-12 Total scores and RV/TLC and RV were not significant; however, those between the D-12 Total scores and FEV_1_ and FEV_1_/FVC were weak but statistically significant (−0.340 and −0.251, respectively). Moreover, the correlation coefficients between the CAT score and airflow limitation, as well as between the CAT score and hyperinflation, were both significant, with the former being the larger, indicating a strong correlation. Similarly, the correlation coefficients between the E-RS Total score and airflow limitation and between the E-RS Total score and hyperinflation were both statistically significant, with the former being larger, suggesting a strong correlation. Therefore, the D-12 Total score, CAT score and E-RS Total score all correlate more strongly with airflow limitation than with hyperinflation.

### 3.3. Relationship between Tools Using the Thresholds

The D-12 Total score was above the threshold in 43 out of the 157 participants (27.4%), and the CAT and E-RS Total scores were higher than the thresholds in 61 (38.8%) and 65 (41.4%), respectively. This result conflicts with the study hypothesis that respiratory symptoms are one of the essential components of health status in subjects with COPD. Therefore, we subsequently analyzed the relationships between tools using a Venn diagram.

The actual number of patients with scores above the threshold is shown in the Venn diagram (Figure 3) (Table 4). Apart from patients who were negative for all three, the largest number were in the location that falls under D-12∩E-RS∩CAT, which included 30 patients (19.1%). The second most common was $\bar{\mathrm{D-12}}$∩E-RS∩CAT, and the third was $\bar{\mathrm{D-12}}$∩$\bar{\mathrm{E-RS}}$∩CAT. If the scores were distributed in complete accordance with the hypothesis, patients should not be distributed in positions $\bar{\mathrm{D-12}}$∩E-RS∩$\bar{\mathrm{CAT}}$, D-12∩E-RS∩$\bar{\mathrm{CAT}}$, D-12∩$\bar{\mathrm{E-RS}}$∩$\bar{\mathrm{CAT}}$ or D-12∩$\bar{\mathrm{E-RS}}$∩CAT. However, in fact, a total of 23 patients (14.6%) fell into these categories. The authors compared the findings of the 134 patients who fitted the hypothesis with those of the 23 who did not but were unable to find any significant difference.

As the analysis using the Venn diagram revealed that the distribution of disability in real cases was not necessarily distributed according to the hypothesis, the next step was to analyze the extent to which the three tools differed in their judgments. Concordant and discordant results between the tools were examined using the threshold (Table 5). The number of those with higher scores on one instrument and lower scores on another was 28 (17.8%) between the CAT and E-RS, 36 (22.9%) between E-RS and D-12 and 42 (26.8%) between the CAT and D-12. If the D-12 score, which suggests the presence of dyspnea, is high, then the CAT score, which suggests impairment of health status, must also be high. However, in real cases, 12 patients (7.6%) had high D-12 scores but normal CAT scores, a seemingly contradictory result. Likewise, the situation in which a lower level concept is disturbed and shows abnormality, but a higher level concept is not disturbed and does not show abnormality, would imply that the hypothesis is inconsistent with the assumption of correctness, which was the case for 16 (10.2%) for the relationship between the CAT and E-RS and for 7 (4.5%) in relation to E-RS and D-12.

## 4. Discussion

This is the first study to examine the hypothesis that in patients with COPD, the relationship between the concepts of dyspnea, respiratory symptoms and health status is that health status is the highest concept, followed by respiratory symptoms and dyspnea as the lowest concepts in a real clinical setting. This hypothesis is supported to some extent by the analysis of the scores of the tools that evaluate PROs designed to evaluate the three concepts. There is, however, some measurement error, as the measuring properties of each tool are naturally related to the distribution of scores. For example, dyspnea is considered an important part of respiratory symptoms, but there were seven patients who had dyspnea but no respiratory symptoms. It would be expected that a patient with a cough and phlegm would be classified as having a problematic health condition, but there were 10 patients who had respiratory symptoms, such as a cough and phlegm, without dyspnea, but whose health condition was not problematic. There were also six patients who had both dyspnea and respiratory symptoms, but whose health status was classified as fine. This distribution of scores contradicting the hypothesis occurs in less than 15% of all cases and might be unavoidable in practical terms. In other words, it can be concluded that the hypothesis was correct in 85% of cases.

In the present study, the Spearman’s correlation coefficients of the scores for health status measured using the CAT, respiratory symptoms using E-RS and dyspnea using D-12 were between 0.6 and 0.7 for any two. In a similar study of a working population of 1566 reported in 2019, the Spearman’s correlation coefficient between D-12 and the CAT was 0.398; between D-12 and E-RS, it was 0.274; and between the CAT and E-RS, it was 0.446 [31], so it can be assumed that the correlation coefficient is quite good in subjects with COPD. However, the associations between dyspnea, respiratory symptoms and health status were significant but far below the level of conceptual similarity. This may be expected as the three PRO measurement tools were created by each developer from independent conceptual frameworks. Therefore, we believe that the three concepts must be distinguished and should not be used as complementary substitutes, especially in a practical clinical setting. In addition, when the correlation coefficients of the CAT, E-RS and D-12 are compared with those of lung volume, airflow limitation, residual volume and diffusion capacity, the correlation coefficients of the first two are very similar, but there seems to be some distance between them and D-12. While we cannot rule out the possibility that this is due to that conceptual factor, we suspect that it is rather related to the fact that the floor effect was quite advanced in D-12. In any case, there are limitations to characterizing the three concepts by studying the distribution of scores and correlations.

We then proceeded with the analysis by classifying each concept into two options, “abnormal” or “normal” or “with disability” or “without,” using a certain threshold value for each. In other words, we attempted to classify patients into (1) with or without dyspnea, (2) with or without respiratory symptoms and (3) with or without impaired health status. However, this yes-or-no format approach is not inherently recommended as it carries the risk of false negatives or false positives. Standardized tools have been developed to avoid such measurement errors. The threshold values used in each tool are also important and may affect the results. For example, in the present study, the CAT threshold used was 10, which GOLD has historically advocated as the boundary between GOLD A and B and between GOLD C and D. However, our 2013 report recommended a threshold of 13.6 [32], and the Canadian Cohort Obstructive Lung Disease (CanCOLD) report by Pinto et al. in 2014 recommended 16 [33]. As our 2019 report was based on 10, we used 10 as the threshold for the CAT in this analysis [31].

Here, it is necessary to verify to what extent it is correct to say that if there is no anomaly regarding one concept, the other concepts are not anomalous either if they are used in a complementary manner in a practical clinical setting. The question is to what extent the two tools agree or disagree on the results of the “abnormal” or “normal” or “with disability” and “without” two-party classification using threshold values. Discordant results were observed in 28 (17.8%) to 42 (26.8%) patients, with the highest number between the CAT and D-12. It is understandable that there were many discrepancies between the most conceptually distant tools, but this may also involve differences in measurement characteristics. However, if the hypothesis is correct here, the condition of a normal D-12 but abnormal CAT is not inconsistent with the hypothesis. As the opposite situation, i.e., an abnormal D-12 and normal CAT, is not allowed by the hypothesis, we can conclude that there was a measurement error in at least 12 (7.6%) of the subjects. Similarly, we believe there was a measurement error between the CAT and E-RS in 16 (10.2%) and between the E-RS and D-12 in 7 (4.5%). The same table is shown in the 2019 report for a working population of 1566 people, where 1~4% of participants showed similar results [31], but the frequency is shown to be higher in the current study for COPD patients.

## 5. Conclusions

In COPD patients, the hypothesis that the relationship between the concepts of dyspnea, respiratory symptoms and health status is that health status is the highest level concept, followed by respiratory symptoms and dyspnea as the lowest level concepts was tested for the first time in a clinical setting. The single correlation coefficients of the scores for health status using the CAT, respiratory symptoms using E-RS and dyspnea using D-12 ranged from 0.6 to 0.7 for any two. The distribution of scores showed a high degree of floor effect in D-12. The correlations with physiological indices were similar for the CAT and E-RS. When each concept was analyzed by categorizing the patients as “abnormal” or “normal” or “with disability” or “without” according to the threshold, 30 patients (19.1%) were considered core cases with dyspnea, respiratory symptoms and impaired health status. The scores were distributed inconsistently, with the hypothesis in less than 15% of the cases; for example, seven patients had dyspnea but no respiratory symptoms. In other words, we can conclude that the hypothesis was supported 85% of the time.

## Figures and Tables

**Figure 1 diagnostics-13-02492-f001:**
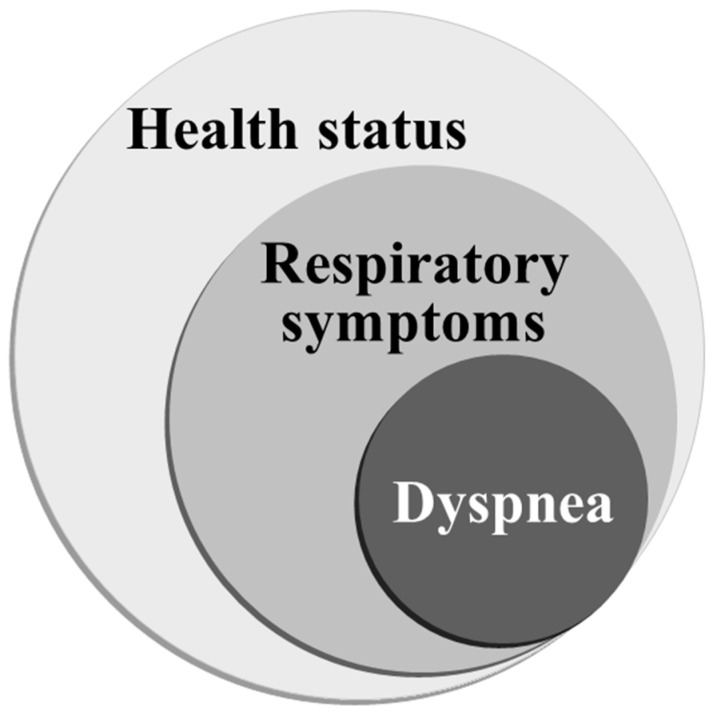
The hypothesis of the present study. Considering health status, respiratory symptoms and dyspnea in subjects with COPD, health status is hypothesized to be the highest concept, and dyspnea is the lowest. We aimed to examine how different they are and to determine whether or not they can be used interchangeably.

**Figure 2 diagnostics-13-02492-f002:**
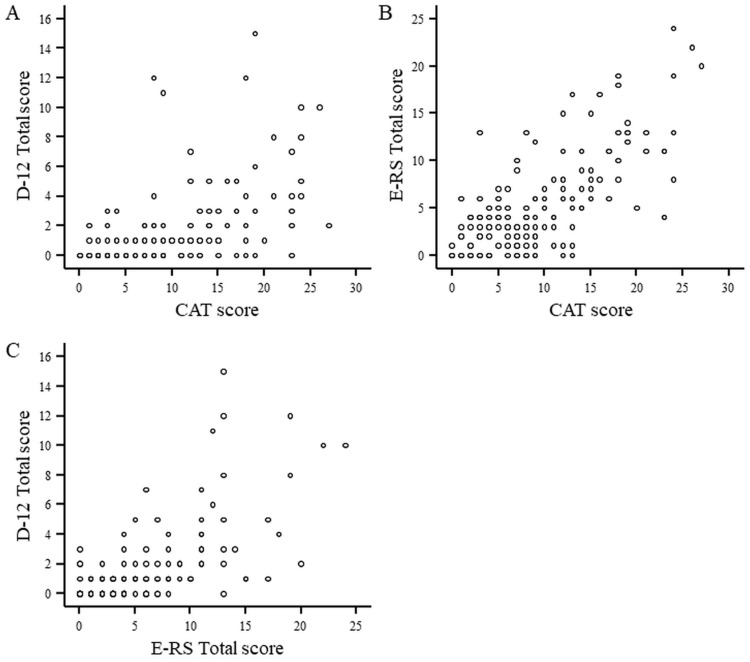
Scatterplots showing the relationships between the D-12 (Dyspnea-12) Total score, CAT (COPD assessment test) score and E-RS (Evaluating Respiratory Symptoms in COPD) Total score in subjects with COPD. (**A**): CAT score vs. D-12 Total score, (**B**): CAT score vs. E-RS Total score, (**C**): E-RS Total score vs. D-12 Total score.

**Figure 3 diagnostics-13-02492-f003:**
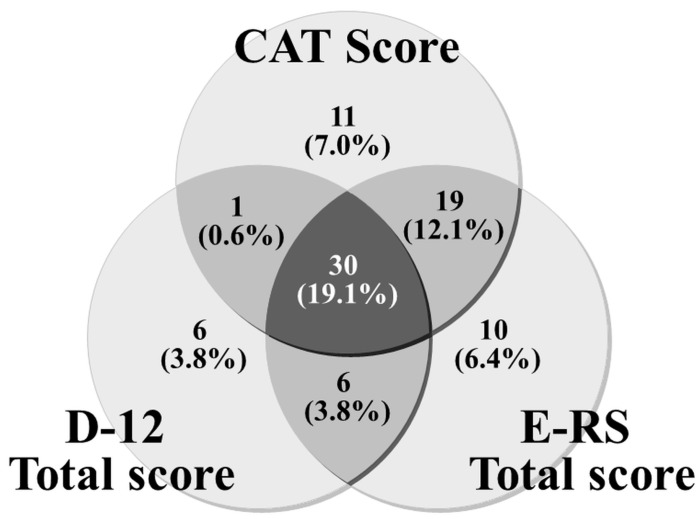
Three-circle Venn diagram representing the relationships between the D-12 Total, CAT and E-RS Total scores. Each reveals the number of positive patients with scores above threshold values.

**Table 1 diagnostics-13-02492-t001:** Baseline characteristics in 157 subjects with COPD and Spearman’s rank correlation coefficients with the scores of the patient-reported outcome measures.

			Mean	SD	Max.	Min.	Correlations (Rs) with		
							D-12 Total Score	CAT Score	E-RS Total Score
Age		years	75.1	6.8	89.0	51.0	—	—	—
BMI		kg/m^2^	22.7	3.3	35.7	14.0	—	—	—
Cumulative Smoking		pack-years	57.7	30.9	204.0	10.0	—	—	—
SVC		% pred.	95.7	18.1	145.9	56.7	−0.250 **	−0.207 **	−0.203 *
FEV_1_		% pred.	69.8	20.4	132.5	21.8	−0.340 ***	−0.335 ***	−0.381 ***
FEV_1_/FVC		%	56.3	10.8	69.9	22.4	−0.251 **	−0.311 ***	−0.391 ***
RV^(1)^		% pred.	125.1	63.7	718.9	28.4	—	0.189 *	0.214 **
RV/TLC ^(1)^		%	44.9	9.7	85.1	18.1	—	0.242 **	0.258 **
DLco ^(2)^		% pred.	53.4	20.6	163.9	8.0	−0.183 *	−0.285 ***	−0.248 **
PaO_2_ ^(3)^		mmHg	79.2	8.9	101.8	56.6	−0.188 *	−0.301 ***	−0.289 ***
SGRQ Total Score		(0–100)	22.8	15.3	63.1	0.9	0.602 ***	0.667 ***	0.636 ***
	SGRQ Symptoms	(0–100)	37.9	20.0	85.3	0.0	0.508 ***	0.549 ***	0.586 ***
	SGRQ Activity	(0–100)	32.1	23.6	87.2	0.0	0.553 ***	0.616 ***	0.578 ***
	SGRQ Impact	(0–100)	13.1	13.1	55.2	0.0	0.533 ***	0.553 ***	0.519 ***
Hyland Scale Score		(0–100)	66.6	16.0	100	20.0	−0.381 ***	−0.513 ***	−0.437 ***
D-12 Total Score		(0–36)	1.5	2.6	15.0	0.0	NA	0.603 ***	0.655 ***
CAT Score		(0–40)	9.1	6.7	27.0	0.0	0.603 ***	NA	0.675 ***
E-RS Total Score		(0–40)	5.2	5.2	24.0	0.0	0.655 ***	0.675 ***	NA

***: *p* < 0.001, **: *p* < 0.01, *: *p* < 0.05; ^(1)^
*n* = 156, ^(2)^
*n* = 154, ^(3)^ one patient receiving oxygen. Missing values of correlation coefficients indicate no statistically significant relationship. D-12, Dyspnea-12; CAT, the COPD Assessment Test; E-RS, Evaluating Respiratory Symptoms in COPD; SGRQ, the St. George’s Respiratory Questionnaire; NA, not available. The numbers in parentheses denote possible score range.

**Table 2 diagnostics-13-02492-t002:** Comparison of patient characteristics and the scores obtained from patient-reported outcomes at baseline by airflow limitation severity in 157 subjects with COPD.

			Stage 1 (*n* = 53)	Stage 2 (*n* = 79)	Stage 3 + 4 (*n* = 25)
			Mean ± SD	Mean ± SD	Mean ± SD
Age		years	75.5 ± 6.6	74.8 ± 7.0	74.7 ± 6.7
BMI		kg/m^2^	23.1 ± 2.8	22.9 ± 3.6	21.4 ± 3.1
Cumulative Smoking		pack-years	47.4 ** ± 23.3	63.5 ± 34.0	61.4 ± 30.5
SVC		% pred.	109.7 *** ± 12.6	92.0 ^§§§^ ± 15.8	77.8 ^¶¶¶^ ± 13.0
FEV_1_		% pred.	91.7 *** ± 10.1	64.9 ^§§§^ ± 9.1	38.7 ^¶¶¶^ ± 8.7
FEV_1_/FVC		%	63.7 *** ± 4.9	56.4 ^§§§^ ± 8.4	40.2 ^¶¶¶^ ± 9.2
RV ^(1)^		% pred.	108.5 ± 31.5	122.0 ^§§§^ ± 47.4	169.3 ^¶¶^ ± 119.0
RV/TLC ^(1)^		%	39.3 *** ± 7.3	45.4 ^§§§^ ± 8.8	54.7 ^¶¶¶^ ± 8.7
DLco ^(2)^		% pred.	58.9 * ± 14.3	53.5 ^§§§^ ± 23.1	41.4 ^¶^ ± 19.3
PaO_2_ ^(3)^		mmHg	81.7 ± 8.9	79.2 ^§§^ ± 8.9	74.4 ^¶^ ± 6.9
SGRQ Total Score		(0–100)	15.0 ** ± 9.9	22.7 ^§§§^ ± 14.3	40.0 ^¶¶¶^ ± 13.9
	SGRQ Symptoms	(0–100)	31.2 ± 16.3	37.0 ^§§§^ ± 19.9	54.9 ^¶¶¶^ ± 17.9
	SGRQ Activity	(0–100)	20.4 ** ± 19.0	31.7 ^§§§^ ± 21.4	58.1 ^¶¶¶^ ± 18.2
	SGRQ Impact	(0–100)	7.1 * ± 7.7	13.3 ^§§§^ ± 12.7	25.1 ^¶¶^ ± 15.1
Hyland Scale Score		(0–100)	72 ± 15	67 ^§§§^ ± 15	54 ^¶¶^ ± 14
D-12 Total Score		(0–36)	0.8 ± 1.8	1.4 ^§§§^ ± 2.3	3.7 ^¶¶^ ± 3.9
CAT Score		(0–40)	7.0 ± 5.4	8.3 ^§§§^ ± 6.1	16.3 ^¶¶¶^ ± 6.7
E-RS Total Score		(0–40)	3.4 ± 3.5	4.8 ^§§§^ ± 5.0	10.2 ^¶¶¶^ ± 6.0

Comparison between Stage 1 and Stage 2 (Steel–Dwass test), ***: *p* < 0.001, **: *p* < 0.01, *: *p* < 0.05; comparison between Stage 2 and Stage 3 + 4 (Steel–Dwass test), ^§§§^: *p* < 0.001, ^§§^: *p* < 0.01,; comparison between Stage 1 and Stage 3 + 4 (Steel–Dwass test), ^¶¶¶^: *p* < 0.001, ^¶¶^: *p* < 0.01, ^¶^: *p* < 0.05; ^(1)^
*n* = 156, ^(2)^
*n* = 154, ^(3)^ one patient receiving oxygen. SGRQ, the St. George’s Respiratory Questionnaire; D-12, Dyspnea-12; CAT, the COPD Assessment Test; E-RS, Evaluating Respiratory Symptoms in COPD. The numbers in parentheses denote possible score range.

**Table 3 diagnostics-13-02492-t003:** Internal consistency and score distribution in the questionnaires.

Patient-Reported Outcomes		Possible Score Range	Items	Score Distribution								
			(*n*)	Mean	SD	Median	75th Percentile	25th Percentile	Max.	Min.	Floor Effect	Ceiling Effect
D-12 Total Score		0–36	12	1.5	2.6	1.0	2.0	0.0	15.0	0.0	47.8%	0%
	D-12 Physical Score	0–21	7	1.4	2.1	1.0	2.0	0.0	10.0	0.0	48.4%	0%
	D-12 Affective Score	0–15	5	0.2	0.9	0.0	0.0	0.0	5.0	0.0	93.6%	0%
CAT Score		0–40	8	9.1	6.7	8.0	13.0	3.0	27.0	0.0	5.1%	0%
E-RS Total Score		0–40	11	5.2	5.2	4.0	8.0	1.0	24.0	0.0	19.7%	0%
	RS—Breathlessness	0–17	5	2.3	3.2	0.0	4.0	0.0	15.0	0.0	50.3%	0%
	RS—Cough and Sputum	0–11	3	1.9	1.7	2.0	3.0	0.0	7.0	0.0	32.5%	0%
	RS—Chest Symptoms	0–12	3	1.0	1.6	0.0	2.0	0.0	6.0	0.0	60.5%	0%

D-12, Dyspnea-12; CAT, COPD Assessment Test; E-RS, Evaluating Respiratory Symptoms in COPD.

**Table 4 diagnostics-13-02492-t004:** Morbidity distribution in the Venn diagram.

					Real-World Clinic *n* (%)	On the Hypothesis
D-12	∩	E-RS	∩	CAT	30 (19.1%)	present, a centrical position
$\bar{\mathrm{D-12}}$	∩	E-RS	∩	CAT	19 (12.1%)	possible
$\bar{\mathrm{D-12}}$	∩	$\bar{\mathrm{E-RS}}$	∩	CAT	11 (7.0%)	possible
$\bar{\mathrm{D-12}}$	∩	E-RS	∩	$\bar{\mathrm{CAT}}$	10 (6.4%)	no
D-12	∩	E-RS	∩	$\bar{\mathrm{CAT}}$	6 (3.8%)	no
D-12	∩	$\bar{\mathrm{E-RS}}$	∩	$\bar{\mathrm{CAT}}$	6 (3.8%)	no
D-12	∩	$\bar{\mathrm{E-RS}}$	∩	CAT	1 (0.6%)	no
$\bar{\mathrm{D-12}}$	∩	$\bar{\mathrm{E-RS}}$	∩	$\bar{\mathrm{CAT}}$	74 (47.1%)	present

D-12, Dyspnea-12; CAT, COPD Assessment Test; E-RS, Evaluating Respiratory Symptoms in COPD.

**Table 5 diagnostics-13-02492-t005:** Concordant and discordant results between tools using the thresholds.

COPD Assessment Test (CAT) and Evaluating Respiratory Symptoms in COPD (E-RS)			
		E-RS Total Score	
		0–4	5 or more
CAT Score	0–9	80 (51.0%)	16 (10.2%)
	10 or more	12 (7.6%)	49 (31.2%)
COPD Assessment Test (CAT) and Dyspnea-12 (D-12)			
		D-12 Total Score	
		0–1	2 or more
CAT Score	0–9	84 (53.5%)	12 (7.6%)
	10 or more	30 (19.1%)	31 (19.7%)
Evaluating Respiratory Symptoms in COPD (E-RS) and Dyspnea-12 (D-12)			
		D-12 Total Score	
		0–1	2 or more
E-RS Total Score	0–4	85 (54.1%)	7 (4.5%)
	5 or more	29 (18.5%)	36 (22.9%)

## Data Availability

Anonymized participant data will be available upon reasonable request to the corresponding author.

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
