# Peer review of "The Conceptual Independence of Health Status, Respiratory Symptoms and Dyspnea in Chronic Obstructive Pulmonary Disease in Real Clinical Practice"

_diagnostics, 2023, doi:10.3390/diagnostics13152492_

Round 1
Reviewer 1 Report
RE: diagnostics-2475814
The Global Initiative for Chronic Obstructive Lung Disease (GOLD) launched a classification system in 2011 stipulates that symptoms should be evaluated using the modified Medical Research Council (mMRC) dyspnea scale or the CAT. However, since the former is regarded as a tool for measuring dyspnea and the latter as a health status measure, the results may differ somewhat from what the symptoms really are. Although some have attributed the discordance to the sensitivity of the tools’ measurement properties, it may not be surprising since dyspnea and health status differ conceptually.
The hypothesis that health status is the highest-ranking concept, followed by respiratory symptoms and dyspnea as the lowest-ranking concepts in subjects with chronic obstructive pulmonary disease (COPD) was tested in a real clinical setting with 157 subjects with stable COPD.
The aim was to ascertain whether the conceptual independence of health status, respiratory symptoms and dyspnea is maintained in the clinical practice of COPD. For this purpose, the authors examined whether the distribution of cases is consistent with the hypothesis when the cases are classified as "abnormal" or "normal" or "with (disability)" or "without (disability)" according to the respective thresholds based on the scores of the evaluation tools for the three concepts.
The authors recruited 157 consecutive patients with stable COPD who attended the outpatient clinic from September 2013 to February 2022. Inclusion criteria included being over 50 years of age, having a smoking history of more than 10 pack-years, having chronic fixed airflow limitation, attending the clinic regularly for more than half a year, having no uncontrolled comorbidities, and having no variation in treatment in the preceding four weeks. Spearman's rank correlation coefficients for scores of health status using the COPD assessment test (CAT), respiratory symptoms using the COPD Evaluating Respiratory Symptoms (E-RS) and dyspnea using the Dyspnoea-12 (D-12) between any two were 0.6 to 0.7.
Upon categorizing the patients as "abnormal" or "normal" according to the threshold, 30 patients (19.1%) had dyspnea, respiratory symptoms, and impaired the health status. Dyspnea was considered an important part of respiratory symptoms though seven patients had dyspnea but no respiratory symptoms. There were 10 patients who had respiratory symptoms without dyspnea but without health status problems. Furthermore, there were six patients who had both dyspnea and respiratory symptoms but whose health status was classified as fine. Thus, the hypothesis was correct in approximately 85% of cases.
The hypothesis that the scores obtained are normally distributed was rejected in the D-12, CAT and E-RS including their subscales. They were skewed toward the milder ends, and a floor effect was observed in all scores. This effect was most pronounced for D-12 (47.8%) and least for the CAT (5.1%). Regarding the interrelationships between the D-12 Total, CAT and E-RS Total scores, they were significantly correlated with each other. All of the correlation coefficients were below 0.7, or what is occasionally regarded as the level suggestive of conceptual equivalence.
Discordant results were observed in 28 (17.8%) to 42 (26.8%) patients, with the highest number between the CAT and D-12. The authors conclude that in COPD patients, health status is the highest-level concept, followed by respiratory symptoms and dyspnea as the lowest level concepts. When each concept was analyzed by categorizing the patients as "abnormal" or "normal" or "with disability" or "without" according to the threshold, 30 patients (19.1%) were considered as core cases with dyspnea, inconsistently with the hypothesis in less than 15% of the cases; in other words, the hypothesis was supported 85% of the time. Finally, since of the 3 measurement tools were developed independently, they should not be used interchangeably in the clinical setting.
General Comments:
The paper is well-written with appropriate sections of introduction, hypothesis, methods, results, conclusions. The references support the findings and arguments the authors make. The lead author is well-published in this area and has used his own previous work to support concepts. In particular, the authors explain how the floor effect may have contributed to the discrepancies between associations amongst concepts (as shown in Fig. 2). Fig. 3 also shows that only 30 of the 157 patients exhibited concordance amongst all 3 tools in which the scores fell above threshold values.
Specific comments:
1. Of 157 patients, only 13 (8.3%) were female – why so? Does this reflect the smoking population of Japan? Or are women generally excluded from such studies?
2. In Table 2, significant differences were found amongst FEV1 and RV and RV/TLC between GOLD stages of COPD. Can the authors conclude whether airways obstruction (represented by FEV1) or air trapping (hyperinflation) represented by RV and RV/TLC correlated more closely with any of the scores for D-12, CAT and E-RS?
Author Response
Author's Reply to the Review Report (Reviewer 1)
- General Comments: The paper is well-written with appropriate sections of introduction, hypothesis, methods, results, conclusions. The references support the findings and arguments the authors make. The lead author is well-published in this area and has used his own previous work to support concepts. In particular, the authors explain how the floor effect may have contributed to the discrepancies between associations amongst concepts (as shown in Fig. 2). Fig. 3 also shows that only 30 of the 157patients exhibited concordance amongst all 3 tools in which the scores fell above threshold values.
Response: Thank you for your careful reading of our manuscript and for your favorable evaluation. We appreciate it very much. We are pleased that many of your comments support our intentions and that we have received a high evaluation. It is a great pleasure to receive such comments, as they are very encouraging for us to carry out our research and publish our paper.
- Specific comments: 1. Of 157 patients, only 13 (8.3%) were female – why so? Does this reflect the smoking population of Japan? Or are women generally excluded from such studies?
Response: We appreciate your comment. Most epidemiological studies in Japan showed that COPD was much more common in men than in western countries. The male to female ratio of COPD patients is thought to be approximately 10 to 1 in Japan. This observation has been widely and generally thought to be attributed to gender differences in the trends in cigarette smoking of the past.
- Specific comments: In Table 2, significant differences were found amongst FEV1 and RV and RV/TLC between GOLD stages of COPD. Can the authors conclude whether airways obstruction (represented by FEV1) or air trapping (hyperinflation) represented by RV and RV/TLC correlated more closely with any of the scores for D-12, CAT and E-RS?
Response: The correlation coefficients between airflow limitation and each PRO measure score, as well as between hyperinflation and each PRO measure score, are presented in Table 1. The correlation coefficients between D-12 Total scores and RV/TLC and RV were not significant, however, those between D-12 Total scores and FEV1 and FEV1/FVC were weak but statistically significant (-0.340 and -0.251, respectively). Moreover, the correlation coefficients between CAT score and airflow limitation, as well as between CAT score and hyperinflation, were both significant, with the former being the larger, indicating a strong correlation. Similarly, the correlation coefficients between E-RS Total score and airflow limitation and between E-RS Total score and hyperinflation were both statistically significant, with the former being larger, suggesting a strong correlation. Therefore, it can be concluded that the D-12 Total score, CAT score, and E-RS Total score all correlate more strongly with airflow limitation than with hyperinflation.
According to your advice, we added the following sentences into the Result section: “The correlation coefficients between baseline characteristics and each PRO measure score are presented in Table 1. The correlation coefficients between D-12 Total scores and RV/TLC and RV were not significant, however, those between D-12 Total scores and FEV1 and FEV1/FVC were weak but statistically significant (-0.340 and -0.251, respectively). Moreover, the correlation coefficients between CAT score and airflow limitation, as well as between CAT score and hyperinflation, were both significant, with the former being the larger, indicating a strong correlation. Similarly, the correlation coefficients between E-RS Total score and airflow limitation and between E-RS Total score and hyperinflation were both statistically significant, with the former being larger, suggesting a strong correlation. Therefore, the D-12 Total score, CAT score, and E-RS Total score all correlate more strongly with airflow limitation than with hyperinflation.” (Lines 174-184).
=============================================
Reviewer 2 Report
This article aimed to test the hypothesis of those hierarchy among dyspnea, respiratory symptom, and health status, finding the validity of the hypothesis 85% in the patients with COPD. The hyerarchy may support the better understanding of the management of COPD patients.
Are there supposed the personalized approach or management in the patients who did not show the accordance to the hypothesis? Is it considered that those patients have distinct pathophysiology?
Author Response
Author's Reply to the Review Report (Reviewer 2)
- Are there supposed the personalized approach or management in the patients who did not show the accordance to the hypothesis? Is it considered that those patients have distinct pathophysiology?
Response: We appreciate your comment. D-12∩∩ or D-12∩∩CAT mean that there were seven patients who had dyspnea but did not have respiratory symptoms. ∩E-RS∩ corresponds to 10 patients who had respiratory symptoms such as cough and phlegm without dyspnea, but whose health condition was not problematic. D-12∩E-RS∩ is six patients who had both dyspnea and respiratory symptoms, but whose health status was classified as fine. They did not show the agreement with the hypothesis. Twenty-three patients (14.6%) fell into these categories. We compared the findings of the 134 patients who fit the hypothesis with those of the 23 who did not but were unable to find any significant difference due to the large bias in the number of cases. The table is included here; it is not included in the revision. Accordingly, the following sentences have been added to the Discussion section of the revision; “The authors compared the findings of the 134 patients who fit the hypothesis with those of the 23 who did not, but were unable to find any significant difference (data not shown).” (Line 209- 211).
  |
  |
  |
  |
(N=134) |
  |
(N=23) |
  |
|
|||||
  |
  |
  |
  |
mean |
  |
SD |
  |
mean |
  |
SD |
  |
|
|
Age |
years |
75.0 |
± |
6.8 |
75.3 |
± |
6.4 |
|
|||||
BMI |
kg/m2 |
22.8 |
± |
3.2 |
22.5 |
± |
3.5 |
|
|||||
Cumulative Smoking |
pack-years |
54.3 |
± |
26.1 |
77.9 |
± |
45.2 |
|
|||||
SVC |
% pred |
95.7 |
± |
18.7 |
95.8 |
± |
14.2 |
|
|||||
FEV1 |
% pred |
70.2 |
± |
21.0 |
67.3 |
± |
15.6 |
|
|||||
FEV1/FVC |
% |
56.5 |
± |
10.9 |
55.2 |
± |
9.9 |
|
|||||
RV1) |
% pred |
127.3 |
± |
67.2 |
112.3 |
± |
33.2 |
|
|||||
RV/TLC1) |
% |
45.1 |
± |
9.7 |
43.2 |
± |
9.5 |
|
|||||
DLco2) |
% pred |
54.3 |
± |
21.1 |
48.5 |
± |
16.1 |
|
|||||
PaO23) |
mmHg |
79.1 |
± |
8.8 |
79.9 |
± |
8.8 |
|
|||||
SGRQ Total Score |
(0 - 100) |
22.6 |
± |
16.0 |
24.3 |
± |
9.2 |
|
|||||
SGRQ Symptoms |
(0 - 100) |
37.4 |
± |
20.7 |
41.0 |
± |
14.1 |
|
|||||
SGRQ Activity |
(0 - 100) |
31.5 |
± |
24.6 |
35.3 |
± |
15.2 |
|
|||||
SGRQ Impact |
(0 - 100) |
13.1 |
± |
13.6 |
13.0 |
± |
9.0 |
|
|||||
Hyland Scale Score |
(0 - 100) |
66.0 |
± |
16.3 |
69.8 |
± |
13.3 |
|
|||||
D-12 Total Score |
(0 - 36) |
  |
1.4 |
± |
2.5 |
  |
2.4 |
± |
3.0 |
  |
|
||
CAT Score |
(0 - 40) |
9.6 |
± |
7.0 |
6.5 |
± |
4.3 |
|
|||||
E-RS Total Score |
(0 - 40) |
5.1 |
± |
5.4 |
5.9 |
± |
3.6 |
|
|||||
1) n=133 vs. 23, 2) n=131 vs. 23, 3) one patient receiving oxygen. SGRQ, the St. George’s Respiratory Questionnaire; D-12, Dyspnoea-12; CAT, the COPD Assessment Test; E-RS, Evaluating Respiratory Symptoms in COPD; The number in a parenthesis means possible score range. |
|
||||||||||||
Round 2
Reviewer 2 Report
All the comments have been appropriately addressed.